# Comparison of Oncologic Outcomes of Dose-Dense Methotrexate, Vinblastine, Doxorubicin, and Cisplatin (ddMVAC) with Gemcitabine and Cisplatin (GC) as Neoadjuvant Chemotherapy for Muscle-Invasive Bladder Cancer: Systematic Review and Meta-Analysis

**DOI:** 10.3390/cancers13112770

**Published:** 2021-06-02

**Authors:** Doo Yong Chung, Dong Hyuk Kang, Jong Won Kim, Jee Soo Ha, Do Kyung Kim, Kang Su Cho

**Affiliations:** 1Department of Urology, Inha University School of Medicine, Incheon 22212, Korea; dychung@inha.ac.kr (D.Y.C.); dhkang@inha.ac.kr (D.H.K.); urostar9988@gmail.com (J.W.K.); 2Department of Urology, Prostate Cancer Center, Gangnam Severance Hospital, Yonsei University College of Medicine, Seoul 06273, Korea; engzsu@yuhs.ac; 3Department of Urology, Soonchunhyang University Seoul Hospital, Soonchunhyang University Medical College, Seoul 04401, Korea; dokyung80@hotmail.com

**Keywords:** bladder cancer, neoadjuvant chemotherapy, gemcitabine, cisplatin, dose-dense MVAC

## Abstract

**Simple Summary:**

Currently, platinum-based neoadjuvant chemotherapy (NAC) is becoming a standard treatment for use in patients with muscle-invasive bladder cancer. However, comparisons of oncologic outcomes for the two most commonly used NAC regimens, ddMVAC (dose-dense methotrexate, vinblastine, doxorubicin, and cisplatin) and GC (gemcitabine and cisplatin), are controversial. We sought to compare the oncologic outcomes of these two regimens via a systematic review and meta-analysis of all the available studies published to date. Through this, we aimed to provide evidence on the optimal NAC regimen for use in muscle-invasive bladder cancer.

**Abstract:**

Platinum-based neoadjuvant chemotherapy (NAC) is widely used for treating muscle-invasive bladder cancer (MIBC). A systematic review was performed following PRISMA guidelines. PubMed, Embase, and the Cochrane Library were searched up to December 2020. We conducted a meta-analysis to compare the oncologic outcomes of ddMVAC (dose-dense methotrexate, vinblastine, doxorubicin, and cisplatin) and GC (gemcitabine and cisplatin), which are the most widely used NAC regimens. Endpoints included pathologic complete response (pCR), pathologic downstaging (pDS), overall survival (OS), and cancer-specific survival (CSS). Five studies, with a total of 1206 patients, were included for meta-analysis. pCR was observed in 35.2% of the ddMVAC arm and in 25.1% of the GC arm, and pCR was significantly higher in ddMVAC than in GC (odds ratio (OR), 1.45; 95% confidence interval (CI), 1.11–1.89; *p* = 0.006). There was no significant difference in pDS (OR, 1.37; CI, 0.84–2.21; *p* = 0.20). OS was significantly higher in ddMVAC than in GC (hazard ratio, 2.16; CI, 1.42–3.29; *p* = 0.0004). Only one study reported CSS outcomes. The results of this analysis indicate that ddMVAC is superior to GC in terms of pCR and OS, suggesting that ddMVAC is more effective than GC in NAC for MIBC. However, this should be interpreted with caution because of the inherent limitations of retrospective studies.

## 1. Introduction

Bladder cancer manifests in most cases as a non-muscle invasive disease and requires only local treatment. Notwithstanding, approximately 25% of bladder cancers invade the muscle layers and 5% have metastatic disease [1]. Radical cystectomy with bilateral pelvic lymph node dissection is a standard local treatment for non-metastatic muscle-invasive bladder cancer (MIBC). However, a large proportion of MIBC patients experience relapse and eventually die after radical cystectomy and pelvic lymph node dissection [2]. Local recurrence rates range from 30% to 54%, and distant relapses occur in up to 50% of cases [3,4,5,6]. Therefore, perioperative chemotherapy, such as adjuvant or neoadjuvant therapy, is used for MIBC. In randomized clinical trials (RCTs) and meta-analyses evaluating the clinical outcomes of neoadjuvant chemotherapy (NAC) [7,8,9], an increase in overall survival (OS) by 5–6% for NAC, compared with radical cystectomy alone, was reported in MIBC patients [10,11].

Various NAC regimens have been tested over the years. The American Urological Association and the European Urological Association guidelines currently recommend platinum-based NAC [3,12]. The most studied platinum-based NACs include methotrexate, vinblastine, doxorubicin, and cisplatin (MVAC) regimens, and gemcitabine and cisplatin (GC) regimens. In 2003, the Southwest Oncology Group-8710 RCT demonstrated that the use of an MVAC regimen for NAC improved survival and pathologic downstaging (pDS) [10]. Another RCT reported that the oncologic outcomes of GC and MVAC regimens were similar; however, the former had a better toxicity profile [13]. One study was conducted on dose-dense MVAC (ddMVAC) plus human granulocyte colony-stimulating factor to supplement the toxicity of MVAC, and a phase 2 trial showed that NAC based on a ddMVAC regimen was well tolerated and safe and that the oncologic results were similar to those of standard regimens [14,15]. Based on these favorable results, ddMVAC and GC have been widely used in NAC in recent years, and the latest National Comprehensive Cancer Network guidelines recommend these two regimens for NAC [12].

However, few studies have compared the two regimens, and a recent phase 3 RCT study did not report long-term follow-up results [16]. Therefore, analyzing studies that have compared ddMVAC and GC regimens for NAC is essential. This systematic review and meta-analysis compares the clinical outcomes of ddMVAC and GC, to determine which is optimal in NAC for patients with MIBC.

## 2. Materials and Methods

### 2.1. Search Strategy and Data Extraction

This systematic review was registered with PROSPERO (CRD42020196422) and complied with the Preferred Reporting Items for Systematic Reviews and Meta-Analyses (PRISMA) statement (http://www.prisma-statement.org/ (assessed on 31 May 2021)) [17]. Relevant studies that compared two NAC regimens (ddMVAC and GC) for MIBC were searched up to December 2020 using PubMed, Ovid-EMBASE, the Cochrane Central Register of Controlled Trials, and the following Medical Subject Headings terms: “bladder cancer”, “bladder carcinoma”, “neoadjuvant”, “MVAC”, “gemcitabine”, “cisplatin”, “regimen”, and relevant variations of these terms. The search was restricted to human studies published in English. Two reviewers (DYC and JWK) independently screened the titles and abstracts of the retrieved articles based on the inclusion criteria. Any discrepancies in the data extracted between the two reviewers were resolved by a third reviewer (KSC). The study was exempt from the approval of an ethics committee or institutional review board because it was a systematic review and meta-analysis.

### 2.2. Inclusion Criteria and Study Eligibility

The eligibility of each study was evaluated taking into account participants, interventions, comparators, outcomes, and study design approach (PICOS): [18] Participants, patients with biopsy-proven MIBC who intended to undergo radical cystectomy and patients who underwent systemic NAC; Interventions, MIBC patients who underwent systemic NAC using ddMVAC; Comparators, MIBC patients who underwent systemic NAC using GC with the same characteristics; Outcomes, comparison of oncologic outcomes (pathologic complete response (pCR), pDS, OS, and cancer-specific survival (CSS)); and Study design, no restrictions on research design, with both randomized controlled studies and nonrandomized observational studies included for analysis.

The primary endpoint was pCR, the secondary endpoint was pDS, and the tertiary endpoints were OS and CSS. Both pCR and pDS were determined by pathological examination after surgery; pDS was defined as decreased pathologic stage compared with the preoperative clinical stage, or downstaging to non-muscle-invasive disease. CSS and OS were defined as the time from the date of surgery to the date of cancer-specific mortality and death from any cause, respectively.

### 2.3. Quality Assessment

A quality assessment was independently performed by two reviewers (DYC and DHK) using the criteria provided by the Cochrane risk-of-bias tool and the Newcastle–Ottawa scale [19,20]. The Cochrane risk-of-bias tool for quality assessments of RCTs was recommended by the Cochrane Handbook for Systematic Reviews of Interventions and includes the following risk-of-bias domains: (1) random sequence generation, (2) allocation concealment, (3) blinding of participants and personnel, (4) blinding of outcome assessment, (5) incomplete outcome data, (6) selective reporting, and (7) other potential biases. Each item was further divided into three categories based on the risk of bias: high, low, and unknown. The three major assessment categories of the Newcastle–Ottawa scale were selection, comparability, and exposure. Studies can be rated up to nine stars. A final score of six stars or more indicates high quality.

### 2.4. Statistical Analysis

Odds ratios (ORs), weighted mean differences, and 95% confidence intervals (CIs) were calculated for dichotomous variables (pCR and pDS). The effects of NAC on OS and CSS were measured using hazard ratios (HRs). Log HR values were obtained from trials reporting HR estimates and CIs, and the standard errors of log HR were calculated using CIs. The effects of ddMVAC and GC on OS and CSS were assessed by pooled HRs and 95% CIs [21].

Between-study heterogeneity was assessed using chi-square and *I*^2^ tests. A Cochran Q statistic *p*-value < 0.05 or *I*^2^ statistic >50% was used to indicate statistically significant heterogeneity between trials [22].

Based on the degree of heterogeneity, either a random-effects or fixed-effects model was applied to calculate summary measures. Data were analyzed using a random-effects model, provided there was evidence of heterogeneity [23]. In the event that at least 10 studies that investigated a particular outcome were included, funnel plots were to be used to assess small effects; however, fewer than 10 studies qualified for this review [24]. The meta-analysis was conducted using Review Manager version 5.3 (RevMan, Copenhagen: The Nordic Cochrane Center, The Cochrane Collaboration, Copenhagen, Denmark, 2013).

## 3. Results

### 3.1. Systematic Review Process

A study selection flowchart according to PRISMA guidelines is presented in Figure 1. The initial database search identified 3317 studies (940 in PubMed, 2121 in OVID-EMBASE, and 256 in Cochran library). Of these, 1457 studies remained for review after removing duplicates. Fifteen articles were excluded after screening the titles and abstracts. Full-text articles were analyzed based on pre-established inclusion criteria. Five studies [16,25,26,27,28], with a total of 1206 patients, were included in the final analysis (Table 1). One study was an RCT, while the others were retrospective case-control studies. Three studies were conducted in the United States, one in the Netherlands, and one in France. All trials enrolled patients diagnosed with MIBC who had undergone either GC or ddMVAC as NAC. 

### 3.2. Quality Assessment

The results of the quality assessment based on the Cochrane risk-of-bias tool are shown in Table 2A. In the RCT, there was a bias of being an unblinded study. Since the schedule of chemotherapy was different, this seemed to be an unavoidable option. The results of the quality assessment using the Newcastle–Ottawa scale for the nonrandomized studies are shown in Table 2B. Four studies received a score of seven points, indicating high quality.

### 3.3. Pathologic Complete Response Rate

Pathologic CR was observed in 35.2% (161/458) of the ddMVAC arm and in 25.1% (188/748) of the GC arm (*p* < 0.001 by Chi-square test) (Table 1). We conducted two analyses, as shown in Figure 2: one with four observational studies, and one with four observational studies and one RCT. In the former analysis (observational studies only), the pCR rate was not significantly different between the two regimens (OR = 1.48; 95% CI, 0.87–2.52; *p* = 0.15), and heterogeneity was found across studies (*I*^2^ statistic, 53%; Cochran Q statistic, *p* = 0.09). In the latter analysis (all studies), the pCR rate was higher in the ddMVAC group (OR = 1.45; 95% CI, 1.11–1.89; *p* = 0.006), and no between-study heterogeneity (*I*^2^ statistic, 43%; Cochran Q statistic, *p* = 0.14) was found.

### 3.4. Pathologic Downstaging Rate

The pDS rates of the ddMVAC and GC regimens were 57.2% (262/458) and 46.5% (348/748), respectively (*p* < 0.001 by Chi-square test) (Table 1). Two analyses were performed as described in item 3.3 (Figure 3). In the former analysis (observational studies only), there were no significant differences in pDS rate between the two regimens (OR = 1.23; 95% CI, 0.62–2.41; *p* = 0.55), and there was heterogeneity across studies (*I*^2^ statistic, 76%; Cochran Q statistic, *p* = 0.005). The latter analysis (all studies) revealed no significant differences between the two regimens (OR = 1.37; 95% CI, 0.84–2.21; *p* = 0.20), and there was some between-study heterogeneity (*I*^2^ statistic, 70%; Cochran Q statistic, *p* = 0.01).

### 3.5. Overall Survival and Cancer-Specific Survival

OS and CSS outcomes between the two regimens are shown in Figure 4. Two studies were included in the OS analysis, and the results indicated that OS was higher in the ddMVAC group versus the GC group (overall HR, 2.16; 95% CI, 1.42–3.29; *p* = 0.0004; *I*^2^ statistic, 0%). Only one study reported CSS outcomes (HR, 2.31; 95% CI, 1.29–4.13; *p* = 0.005).

## 4. Discussion

Since the Southwest Oncology Group reported a positive effect of NAC using MVAC for MIBC in 2003 [10], NAC before radical cystectomy and pelvic lymph node dissection has been used as the standard treatment for MIBC. A few years later, GC and ddMVAC regimens were established as the standard NAC treatments, with low morbidity, low toxicity, and good oncologic outcomes [12,13,14,29].

Various NAC regimens have been implemented in the past few years, and clinical trials on immune-oncology agents are underway [30,31,32]. Although studies on novel NAC treatments are ongoing, large-scale RCTs and long-term follow-up studies are lacking. Therefore, finding and evaluating optimal platinum-based NAC regimens for cisplatin-eligible MIBC patients is crucial.

A phase 3 RCT reported that ddMVAC caused more severe asthenia and gastrointestinal side effects than GC in perioperative chemotherapy, but elicited a significantly higher local control rate (pCR, pDR, or organ-confined tumors) in MICB patients [16]. However, this RCT has not yet reported long-term oncologic outcomes, such as OS, CSS, and progression-free survival [16]. In this respect, the present study helped identify an optimal platinum-based NAC regimen. Our meta-analysis demonstrated that ddMVAC was superior to GC with regards to pCR and OS. In addition, considering that there was no detectable difference in toxicity profiles and tolerability between ddMVAC and GC [14,15,33], ddMVAC should be considered the standard of care for MIBC [14,15,33].

Our analysis of two retrospective studies showed that OS was better in the ddMVAC group. However, insufficient data on the long-term comparative oncological outcomes, such as OS, CSS, and progression-free survival, between ddMVAC and GC is a limitation. It is expected that an answer will be obtained through follow-up results from ongoing phase 3 RCTs. Meanwhile, we believe that the long-term oncological results will be better in the ddMVAC arm because ddMVAC is superior to GC regarding pCR. Previous studies reported that the prognosis in patients with pCR after NAC was good. Petrelli et al. conducted a meta-analysis to determine whether pCR after NAC was associated with an improved outcome in MIBC [29] and found that patients with pCR after NAC and radical cystectomy had a 55% lower risk of mortality (relative risk (RR), 0.45; 95% CI, 0.36–0.56; *p* < 0.00001) and an 81% lower risk of recurrence, compared with patients with pathologic residual disease (RR, 0.19; 95% CI, 0.09–0.39; *p* < 0.00001). A recent cohort study enrolled 1553 patients (314 with pCR and 1239 with pathologic residual disease) and found that patients with pCR had better OS than those without pCR and that the average HR for pathologic residual disease versus pCR was 4.56 (95% CI, 3.34–6.26) [34], suggesting that pCR following NAC improves survival in MIBC.

Nevertheless, there is controversy regarding the optimal number of cycles of ddMVAC in a neoadjuvant setting. No studies have compared the optimal cycles of ddMVAC. The results of four retrospective studies showed that pCR and pDS were 21.0–41.3% and 37.5–69.0%, respectively, after three to four cycles of ddMVAC [25,26,27,28] whereas, the RCT found that pCR and pDS were 42.2% and 63.3%, respectively, after a six-cycle course [16]. However, a six-cycle course may increase side effects, potentially delaying surgery. In addition, the longer the time period between NAC and radical cystectomy, the more the cancer progresses [35]. Therefore, improvement in pCR and pDS after six cycles of ddMVAC does not necessarily improve survival, and further research is needed to determine the ideal number of cycles of ddMVAC.

This study has limitations. First, the number of selected studies was small because few studies in the literature have compared the two regimens. Second, four studies were retrospective and were, therefore, prone to biases related to treatment allocation, grouping, and data collection. Notwithstanding, the retrospective studies included in our study were of high quality when evaluated using the Newcastle–Ottawa scale. In addition, there were no studies showing significant differences between the two groups based on a table comparing the baseline characteristics of the ddMVAC group and the GC group in each study. Third, the results of OS and CSS should be interpreted with caution because of the small number of long-term follow-up studies. Despite these limitations, this study is the first meta-analysis to compare oncologic outcomes between ddMVAC and GC for NAC. We provide evidence that ddMVAC may be a better NAC treatment than GC in patients with MIBC. Additional well-designed RCTs are necessary to confirm our conclusion.

Currently, studies on NAC responses in patients with variant histology or expression of specific biomarkers are in progress [36,37,38]. For instance, Miron et al. has reported that patients with mutations in *ATM*, *RB1*, or *FANCC* had a better response to NAC and improved long-term survival [37]. As mentioned earlier, NAC studies on immunotherapy are underway. In NAC for MIBC using immunotherapy, studies including ABACUS trial (atezolizumab) [32,39] and PURE-01 (pembrolizumab) [30] have been conducted. Although they have not yet published a large-scale phase 3 RCT study, they have shown good results in preliminary studies. In these studies, a patient’s pathologic response was found to be related to biomarker results. Although large-scale studies should be published in the future, these immunotherapy results may serve as the basis for individual, patient-specific treatments.

## 5. Conclusions

In our meta-analysis, ddMVAC was superior to GC with regards to pCR and OS, suggesting that ddMVAC is more effective than GC in NAC for MIBC. However, this finding should be interpreted with caution because of the inherent limitations of retrospective studies. Large-scale RCTs and long-term follow-up studies are warranted to validate these outcomes.

## Figures and Tables

**Figure 1 cancers-13-02770-f001:**
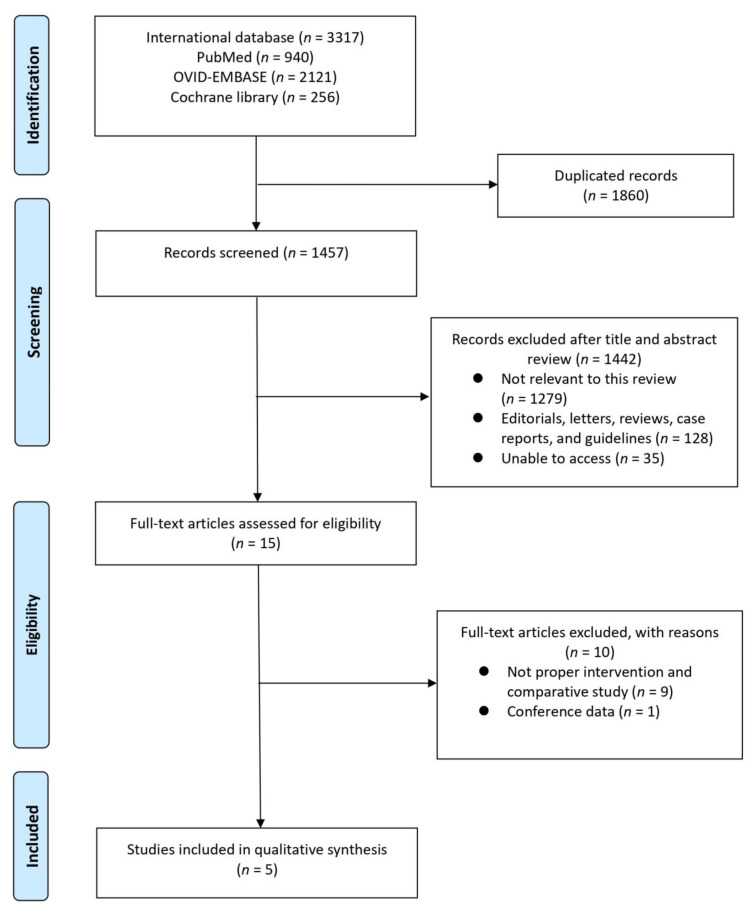
Study selection flowchart according to the Preferred Reporting Items for Systematic Reviews and Meta-analysis guidelines.

**Figure 2 cancers-13-02770-f002:**
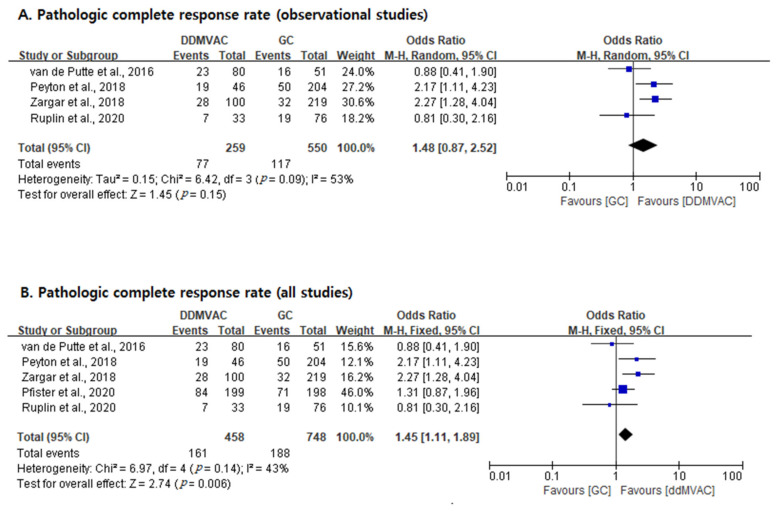
Forest plots of pathologic complete response rates. (**A**): Observational studies only. (**B**): All studies.

**Figure 3 cancers-13-02770-f003:**
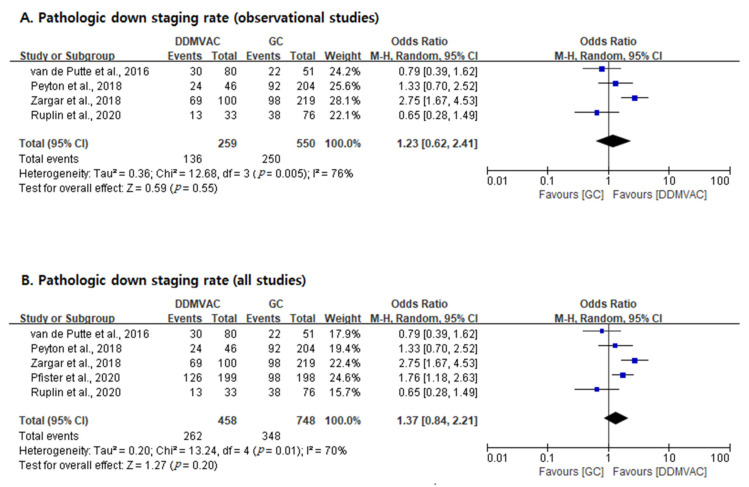
Forest plots of pathologic downstaging rates. (**A**): Observational studies only. (**B**): All studies.

**Figure 4 cancers-13-02770-f004:**
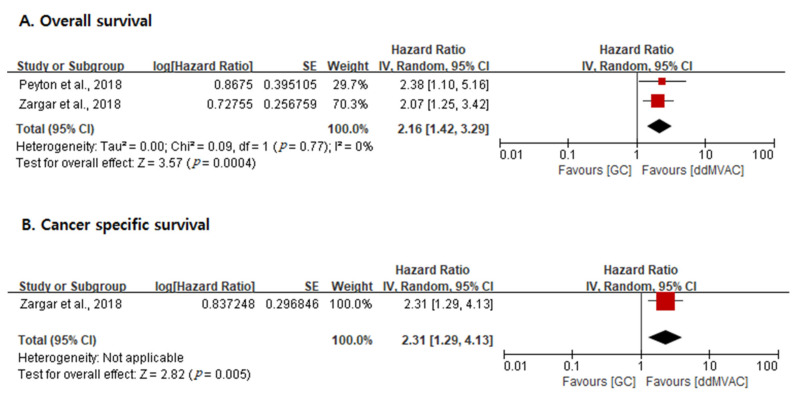
Forest plots of survival outcomes. (**A**): Overall survival. (**B**): Cancer-specific survival.

**Table 1 cancers-13-02770-t001:** Characteristics of the eligible studies.

Authors Year Country	Study Design	Study Summary	NAC Regimen	Total Patients	pCR		pDS		OS HR (95% CI) *p* Value	CSS HR (95% CI, *p* Value)	Median Follow Up (95% CI or IQR)
					No (%)	*p* Value	No (%)	*p* Value			
Van de Putte et al. [25] 2016 Netherlands	Retrospective single-institutional	Comparisons of oncologic outcomes between ddMVAC and GC or classic MVAC as NAC for ≥cT2 MIBC. * This meta-analysis only analyzed comparitive data between ddMVAC and GC, excluding other data.	ddMVAC 4 cycle	80	23 (28.75)	0.845	30 (37.50)	NR	NR	NR	NR
			GC 4 cycle	51	16 (31.37)		22 (43.14)		NR	NR	NR
Peyton et al. [26] 2018 USA	Retrospective, single-institutional	Comparisons of oncologic outcomes between ddMVAC and GC as NAC for ≥cT2 MIBC.	ddMVAC 3–4 cycle	46	19 (41.30)	<0.001	24 (52.17)	0.02	0.42 (0.17–1.06) *p* = 0.07	NR	13.8 months (12.3–16.1)
			GC 3–4 cycle	204	50 (24.51)		92 (45.10)		1	NR	
Zargar et al. [27] 2018 USA	Retrospective, multi-institutional	Comparisons of oncologic outcomes between ddMVAC and GC as NAC for ≥cT3 MIBC.	ddMVAC 3–4 cycle	100	28 (28.00)	0.01	69 (69.00)	0.08	1	1	1.8 years (IQR 0.5–4.1)
			GC 3–4 cycle	219	32 (14.61)		98 (44.75)		2.07 (1.25–3.42) *p* = 0.005	2.31 (1.29–4.13) *p* = 0.005	1.2 years (IQR 0.5–2.9)
Pfister et al. [16] 2020 France	Prospective, multi-institutional	Comparisons of oncologic outcomes between ddMVAC and GC as NAC for ≥cT2 MIBC.	ddMVAC 6 cycle	199	84 (42.21)	0.021	126 (63.32)	0.007	NR	NR	NR
			GC 4 cycle	198	71 (35.86)		98 (49.49)		NR	NR	
Ruplin et al. [28] 2020 USA	Retrospective, single-institutional	Comparisons of oncologic outcomes between ddMVAC and GC or switch regimen as NAC for ≥cT2 MIBC. * This meta-analysis only analyzed comparitive data between ddMVAC and GC, excluding other data.	ddMVAC 3–4 cycle	33	7 (21.21)	0.67	13 (39.39)	0.31	NR	NR	NR
			GC 3–4 cycle	76	19 (25.00)		38 (50.50)		NR	NR	NR

CI, confidence intervals; CSS, cancer-specific survival; ddMVAC, a dose-dense combination of methotrexate, vinblastine, doxorubicin, and cisplatin; GC, gemcitabine, cisplatin; IQR, interquartile range; MIBC, muscle-invasive bladder cancer; NAC, neoadjuvant chemotherapy; NR, not reported; OS, overall survival; pCR, pathologic complete response; pDS, pathologic down-staging rate.

**Table 2 cancers-13-02770-t002:** Results of quality assessment using the Cochrane risk-of-bias tool and Newcastle–Ottawa scale.

**A. Results of Quality Assessment of the Randomized Control Trial Using the Cochrane Risk-of-Bias Tool**									
Author(s) (Year)	Random Sequence Generation (Selection Bias)	Allocation Concealment (Selection Bias)	Blinding of Participants and Personnel (Performance Bias)		Blinding of Outcome Assessment (Detection Bias)	Incomplete Outcome Data Addressed (Attrition Bias)	Selective Reporting (Reporting Bias)		Other bias
Pfister et al. [16] (2020)	Low risk	Low risk	High risk		High risk	Low risk	Low risk		Unclear
**B. Results of Quality Assessment of Nonrandomized Studies Using the Newcastle–Ottawa Scale**									
Author(s) (Year)	Selection (4)				Comparability (2)	Exposure (3)			Total score
	Adequate definition of cases	Representativeness of cases	Selection of controls	Definition of controls	Control for important factor or additional factor	Ascertainment of exposure	Same method of ascertainment for cases and controls	Non-Response rate	
Van de Putte et al. [25] (2016)	1	1	0	1	2	1	1	0	7
Peyton et al. [26] (2018)	1	1	0	1	2	1	1	0	7
Zargar et al. [27] (2018)	1	1	0	1	2	1	1	0	7
Ruplin et al. [28] (2020)	1	1	0	1	2	1	1	0	7

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
