# Peer review of "Comparison of Oncologic Outcomes of Dose-Dense Methotrexate, Vinblastine, Doxorubicin, and Cisplatin (ddMVAC) with Gemcitabine and Cisplatin (GC) as Neoadjuvant Chemotherapy for Muscle-Invasive Bladder Cancer: Systematic Review and Meta-Analysis"

_cancers, 2021, doi:10.3390/cancers13112770_

Round 1

Reviewer 1 Report

The authors conducted a systematic review and meta-analysis on ddMVAC and GC therapies in the neoadjuvant setting before radical cystectomy. The included four retrospective clinical studies as well as one randomised clinical trial and showed that ddMVAC is superior to GC in terms of pathologic complete response, pathologic downstaging, overall survival and cancer-specific survival. Although the reviewer agrees that ddMVAC would be beneficial in certain settings, there are several issues that should be addressed before considered for publication.

Major points

  1. This systematic review includes a relatively small number of studies of which most are retrospective studies. Therefore, possible biases must be taken into consideration. As the authors themselves mentioned, ddMVAC would cause more adverse events than GC, the selection bias towards selecting healthier patients in the ddMVAC arm is likely to exist, leading to favourable outcomes in patients receiving ddMVAC. Can the authors eliminate this possibility?
  2. Moreover, publication biases should also be considered. A Funnel plot might as well be presented in order to assess this important factor that could significantly skew the interpretation of the results.

Minor points

  1. Incomplete sentence in Lines 165-166: ‘the pCR rate was no significant differences in the ddMVAC arm’
  2. A comma is lacking in a line 212 ‘OS, CSS(,) and PFS’

Reviewer 2 Report

This is a well conducted meta-analysis regarding the efficacy of ddMVAC and GC as neoadjuvant chemotherapy for muscle-invasive bladder Cancer.

There are some comments:

  1. the ddMAV is report to some advantage compared to G+C in metastatic UC, please describe the main purpose of this study, focus on neoadjuvant setting for muscle-invasice baldder cancer?
  2.  does these finding can be apply to Upper urinary tract urothelial cancer ( ureteral cancer and renal pelvic cancer)? 
  3. does these finding also the same for metastatic bladder cancer? like for lymph node positive and distant metastaisis bladder cancer
  4. please discuss the role of concurrent radiotherapy in this setting
  5. please discuss the role of ddMAV vs. G+C for muscle invasive bladder cancer in NCCN guidelines / AUA /other guidelines
  6. in modern era, how about the role of immuno-therapy for muscle invasive bladder cancer, this issue should also discussed
  7. what the novelty of this study, the role of chemotherapy for muscle -invasive bladder cancer remain an important issue? 

Reviewer 3 Report

Methodology of the systematic review/metanalysis is overall correct. The main limitation remains the relatively high risk of bias of the four retrospective studies included (even if they may be classified of high quality). They account for most of the patients included. Authors correctly reported the limitation, but it should be more extensively discussed to clarify the meaning of their findings 

Round 2

Reviewer 1 Report

All the issues raised in the first review process have been resolved.

Reviewer 2 Report

The raised questions have been answered adequately.